

# Smartphone-based reminder system to promote pelvic floor muscle training for the management of postnatal urinary incontinence: historical control study with propensity score-matched analysis

Kaori Kinouchi and Kazutomo Ohashi

Department of Children and Women's Health, Osaka University Graduate School of Medicine, Suita, Osaka, Japan

## ABSTRACT

**Background**. The purpose of this study was to evaluate the efficacy of a smartphone-based reminder system in promoting pelvic floor muscle training (PFMT) to help postpartum women manage urinary incontinence (UI).

**Methods**. Forty-nine and 212 postpartum women in the intervention and control groups, respectively, received PFMT guidance using a leaflet and verbal instruction as the standard care at an obstetrics clinic in Japan. Women in the intervention group also received PFMT support using the smartphone-based reminder system between January and August 2014. For analysis, they were compared with historical controls between February 2011 and January 2012, who did not receive such support and were chosen by propensity score matching. The outcomes examined were PFMT adherence and UI prevalence. The former consisted of implementation rate (i.e., the percentage of women who reported performing PFMT during the intervention period), training intensity (i.e., the number of pelvic floor muscle contractions (PFMCs) per day), and training frequency (i.e., the number of days PFMT was performed per week); the latter consisted of self-reported UI prevalence at baseline and at the end of the eight-week intervention period.

**Result**. Propensity score matching resulted in 58 postpartum women ($n = 29$ per group). The intervention group exhibited better PFMT adherence than the control group, in terms of PFMT implementation rate (69 vs. 31%, $p = 0.008$), median training intensity (15 vs. 1 PFMC reps/day, $p = 0.006$), and training frequency (7 vs. 3 days/week, $p < 0.001$). UI prevalence was not different between the groups at baseline, but was significantly reduced in the intervention group at eight weeks (0 vs. 24%, $p = 0.004$).

**Conclusion**. Our smartphone-based reminder system appears promising in enhancing PFMT adherence and managing postpartum UI in postpartum women. By enhancing PFMT adherence and improving women's ability to manage the condition, the reminder system could improve the health-related quality of life of postpartum women with UI.

Corresponding author
Kaori Kinouchi,
kinouchi@sahs.med.osaka-u.ac.jp

## INTRODUCTION

The International Continence Society (ICS) defines urinary incontinence (UI) as the "involuntary loss of urine." In a population-based study of countries with response rates >60%, the reported prevalence of UI among women was 2.8–46.3% (*Abrams et al., 2012*). An epidemiological survey of Japanese women aged ≥40 years revealed that 23% experienced UI at least once per week (*Homma et al., 2006*). Pregnancy and childbirth are risk factors for UI (*Wesnes et al., 2007*; *Wesnes et al., 2010*). The condition affects 20–35% of primiparous women following vaginal delivery between the first and third months postpartum (*Borello-France et al., 2006*; *Ekstrom et al., 2008*; *Dinc, Kizilkaya & Yalcin, 2009*). While it is considered a minor symptom in postpartum women, it certainly could not be disregarded. UI is a major problem that lowers health-related quality of life (HRQOL) (*Goldberg et al., 2005*) and may cause both social and hygiene problems.

Pelvic floor muscle training (PFMT) is considered the first-line conservative treatment for UI in clinical practice guidelines in Japan (*MINDS, 2013*), the UK (*NICE, 2015*) and the USA (*NGC, 2006*). The approach has been proven effective at treating and preventing postpartum UI (*Dumoulin et al., 2015b*). However, the efficacy of PFMT commencing specifically postnatally to treat UI was only measured in three randomized controlled trials (RCTs), of which two were small-scale studies of under 20 participants. Thus, further research is necessary.

Intensive PFMT, in which women are taught the exercises by physical therapists or specialists and receive periodic follow-up on training frequency and efficacy, is reportedly better at improving UI symptoms (*Dumoulin et al., 2015a*). However, this approach is associated with high costs; patients have to visit the hospital or clinic for follow-up or medical staff have to visit them at home to provide guidance, and the approach is not covered by the healthcare systems of several countries, including Japan. Moreover, one study involving Denmark women (mean age = 54 years) found that although patients are instructed in intensive PFMT programs by obstetricians, gynecologists, or urologists to adhere to the protocol to achieve the best results, <50% visited the hospital for at least two-thirds of the scheduled follow-up appointments (*Tibaek & Dehlendorff, 2013*).

Postpartum women, whose time may be otherwise occupied by childcare, would likely have more difficulty in adhering to the protocols than would the women described in the aforementioned study. However, few studies have investigated PFMT adherence in this population. Moreover, postpartum women who lack the time to commit to intensive PFMT programs need a simple yet free way to manage UI. To address this, we developed a smartphone-based reminder system to provide PFMT support at home and enhance their adherence, thereby eliminating the need for follow-up visits to their healthcare provider as in conventional intensive programs.

In this study, we sought to clarify PFMT adherence in postpartum women and determine whether a smartphone-based reminder system could help them manage UI. Our objective was to assess the effectiveness of our PFMT support smartphone-based reminder system in helping women manage postpartum UI.

## MATERIALS & METHODS

### Study design

In this case-control study, postpartum women in the smartphone group, who received PFMT support via a smartphone-based reminder system, were compared with a historical control group, who did not use the reminder system. All participants received typical PFMT education administered by a midwife using a leaflet and verbal instruction.

### PFMT support using smartphone-based reminder system

To develop the smartphone-based reminder system, we used the Rtime$^{TM}$ system (NEUES Co., Japan) (*Higashida et al., 2015*), which is designed to help patients follow medication and hospital visit schedules and works on iOS$^{TM}$ or Android$^{TM}$ smartphone platforms. Approximately 950,000 yen (8,400 USD) was used to develop the system, and approximately 450,000 yen (4,000 USD) is needed for the maintenance. A Grant-in-Aid helped us pay the costs. To register their smartphones with the system, the participants used one of two ways that involved sending blank e-mails. After a blank e-mail was sent, the system sent a response with a URL that directed them to an initial registration webpage and another URL for a website where they were instructed to enter their background characteristics (age, body mass index (BMI) before pregnancy, weight gain during pregnancy, and their child's birth weight). Data were encrypted using SSL during sending and receiving to prevent leakage of personal information; the technology is commonly used by websites that handle personal or credit card information. The day after the initial registration, the system started sending PFMT reminder messages every day at 9 a.m. The reminder displayed the number of days that had passed since the start of the training and a message with an emoji encouraging the participants to train (''Remember your PFMT today!'') (Fig. 1A). On the first day, the message also displayed a URL where the participants were asked to enter their current UI status by answering the following question: ''Do you have involuntary loss of urine that is a social or hygiene problem?'', which is based on the ICS definition at the time (*Abrams et al., 2012*); the system sent e-mails every day for eight weeks, i.e., a total of 56 reminder messages. The day after the final reminder message, the system automatically sent an e-mail with a URL where the participants were asked to enter PFMT adherence-related information and current UI status. The websites for entering data were designed with pull-down menus and radio buttons so that the participants could easily enter the needed information (Fig. 1B). No participants withdrew their participation because of difficulty in using the system during the study period. All data that the participants entered on the websites were stored on the system's server. The data can be accessed by logging into a system management website; login IDs and passwords were managed only by the researchers.

### Participants

*Smartphone group*

Participants for the smartphone group were recruited from January to August 2014 at an obstetric clinic in Osaka Prefecture, Japan, which performs approximately 600 deliveries per year. Posters explaining the study were displayed inside the clinic, and women who

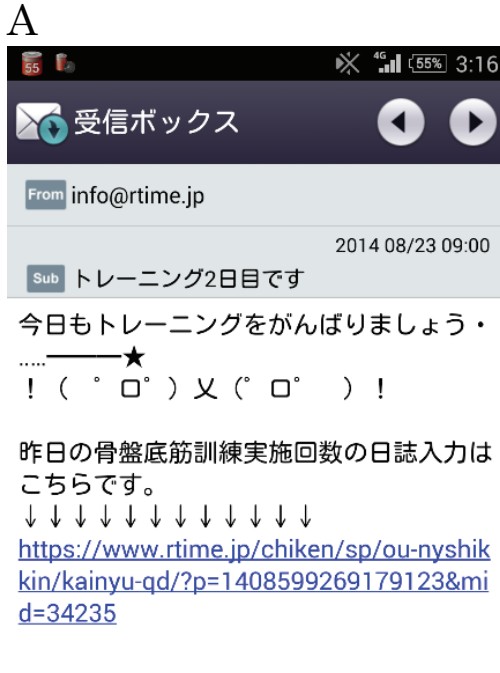
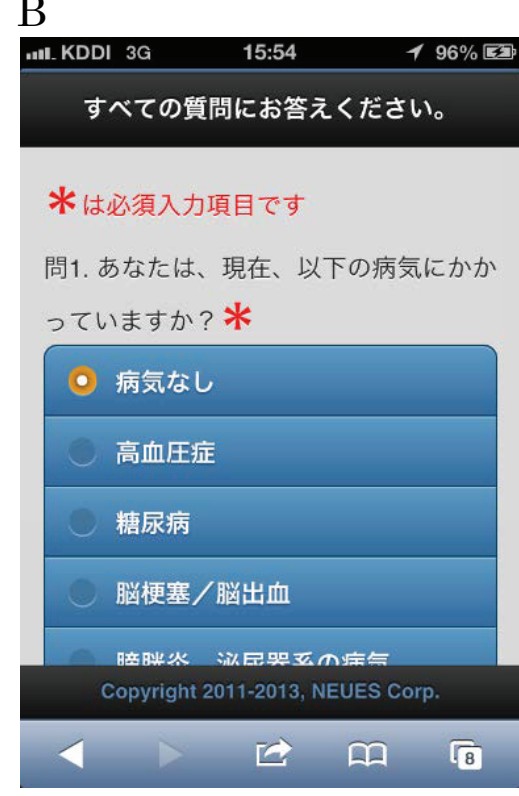

**Figure 1** **Screenshots of PFMT reminder e-mail and input screen for participant characteristics.** (A) Screenshot of PFMT reminder e-mail. (B) Screenshot of input screen for participant characteristics.

visited the clinic for 1-month postpartum examinations were invited to participate. The inclusion criteria included having undergone vaginal delivery and having a smartphone. The exclusion criteria were history of pelvic surgery and cerebral infarction, as well as current hypertension, diabetes, hemorrhage, cystitis, neurological disease of the urinary system, chronic cough, and diuretic use. Women were recruited in the clinic's waiting room during the 20- to 30-min wait for a 1-month postpartum checkup. After confirming that the postpartum women had a smartphone and fulfilled the other criteria, the study was explained verbally and in writing. A midwife, who was either part of the research or clinic staff, explained to the postpartum women who agreed to participate how to register with the system and also provided an explanation on how to accurately perform PFMT using a leaflet.

After registering, the women began receiving PFMT reminder messages via the smartphone-based reminder system. Eight weeks thereafter, they were sent an e-mail with a link to a website with a questionnaire. Aside from answering the question "Did you do PFMT these eight weeks?" they were asked to self-report their training intensity (i.e., the number of pelvic floor muscle contractions (PFMCs) they performed per day; PFMC reps/day), training frequency (i.e., the number of days they performed PFMT per week; PFMT days/week), and their UI status.

### Control group

Women in the historical control group had given birth at the same obstetric clinic and were asked to participate in a previous, different research concerning postpartum PFMT adherence and UI status between February 2011 and January 2012. The inclusion criterion was postpartum women who had vaginal delivery; the exclusion criteria were the same as those in the smartphone group, history of pelvic surgery and cerebral infarction, as well as current hypertension, diabetes, hemorrhage, cystitis, neurological disease of the urinary system, chronic cough, and diuretic use. Midwives employed by clinic explained the research (verbally and in writing), distributed the questionnaire, and provided the same PFMT instruction as that in the smartphone group. The participants who agreed to participate answered a similar initial questionnaire as the smartphone group regarding their background characteristics and current UI status. After eight weeks, another questionnaire was sent by mail with the same question items as those in the smartphone group, which asked the participants to self-report their training, intensity, frequency, and current UI status.

### Pelvic floor muscle training

Women in both the smartphone and control groups received the same training: a midwife used a leaflet and provided instruction on how to perform PFMT. The leaflet contained information on the functions of the female pelvic floor muscle group and postpartum UI and described five essential points when performing PFMT. Using a sagittal cross-section diagram of the female pelvic area, the midwife explained how the pelvic floor muscle group supports the pelvic organs and prevents UI from occurring when functioning normally. Subsequently, she explained how the pelvic floor muscle group is damaged by pregnancy and childbirth and how incontinence can occur if the damage is unaddressed. In addition, PFMT included the following five points: (1) relax the body when performing the training, (2) slowly clench the vagina and anus as strongly as possible without flexing the abdomen, (3) keep the pelvic floor muscles clenched while counting up to 3–6, (4) perform three sets of six contractions every day (a total of 18 per day), and (5) continue the training for at least eight weeks. A diagram explaining the postures to be used during training (standing, supine with the knees bent, prone with the elbows and knees on the floor, and standing while leaning on a table) was also employed. The method was based on Chiarelli's PFMT method (*Chiarelli, Murphy & Cockburn, 2004*), which has been deemed effective for antenatal and postpartum women in a systematic review (*Boyle et al., 2012*).

### Measurement

Participants provided the following information: age, BMI before pregnancy, weight gain during pregnancy, and their child's birth weight. The participants referred to their maternal handbooks for the latter two. Women in the intervention group provided these data via a website after completing their initial registration in the smartphone-based reminder system. Women in the control group wrote this information on the first questionnaire, which assessed the women's background characteristics and UI status; after eight weeks, they were sent another questionnaire with the same question items as those in the smartphone group.

Our adherence outcomes included PFMT implementation rate (calculated as the percentage of postpartum women who responded that they performed PFMT during the eight-week intervention period) and median training intensity and frequency. UI prevalence at baseline and at eight weeks was assessed by the following question: "Do you have involuntary loss of urine that is a social or hygiene problem?", and was calculated as the percentage of women answering "yes" to the question. Women in the smartphone group responded via the website; women in the control group responded through the questionnaire at baseline and eight weeks thereafter.

## Ethical considerations

The participants provided consent to participate in the study and to have their data published after the objectives, methods, expected results, research cooperation benefits, and disadvantages were explained to them. We also explained that not participating would not put the participant at a disadvantage to ensure that their cooperation is not forced. Moreover, individuals in the smartphone group provided a written consent to participate in the study. Agreement of women in the control group to participate was confirmed by the answered questionnaire. This research was approved by Osaka University Medical Science Department of Health Ethics Committee (approval number: 151, 268).

## Statistical analysis

Propensity scores were calculated to adjust for the background characteristics of the smartphone and control groups, which was followed by matching. Age, parity, and BMI (each being considered a risk factor for UI) were considered predictor variables to calculate propensity scores. Match tolerance was set as perfect agreement.

Continuous variables are expressed as median values with interquartile range; categorical variables are expressed as number of cases with percentage. Background characteristics of the smartphone group were compared with those of the control group using the Mann–Whitney U-test for continuous variables and Fisher's exact test for categorical variables. Median values for PFMC reps/day and PFMT days/week were calculated and tested after Bonferroni correction for multiple comparisons. UI prevalence was directly evaluated using binomial distribution, as the only possible values were 0 and 1. All tests were two-sided, with $p < 0.05$ considered to indicate statistical significance. Statistical analysis was conducted using SPSS version 24 (IBM Japan, Inc.).

## RESULTS

Figure 2 showed flow diagram of postpartum women participating in the study. Forty-nine women who met the inclusion criteria were included in the smartphone group. However, two women did not record their current UI status the day after the initial registration and 16 women did not after eight weeks, respectively, and thus were excluded. Data of the remaining 31 women were analyzed. The control group initially consisted of 212 women who returned the initial questionnaire. However, eight were excluded for having had a cesarean section, one for history of cystitis, seven for incomplete responses, and 75 for failing to return the second questionnaire at eight weeks. Thus, data of 121 women in the

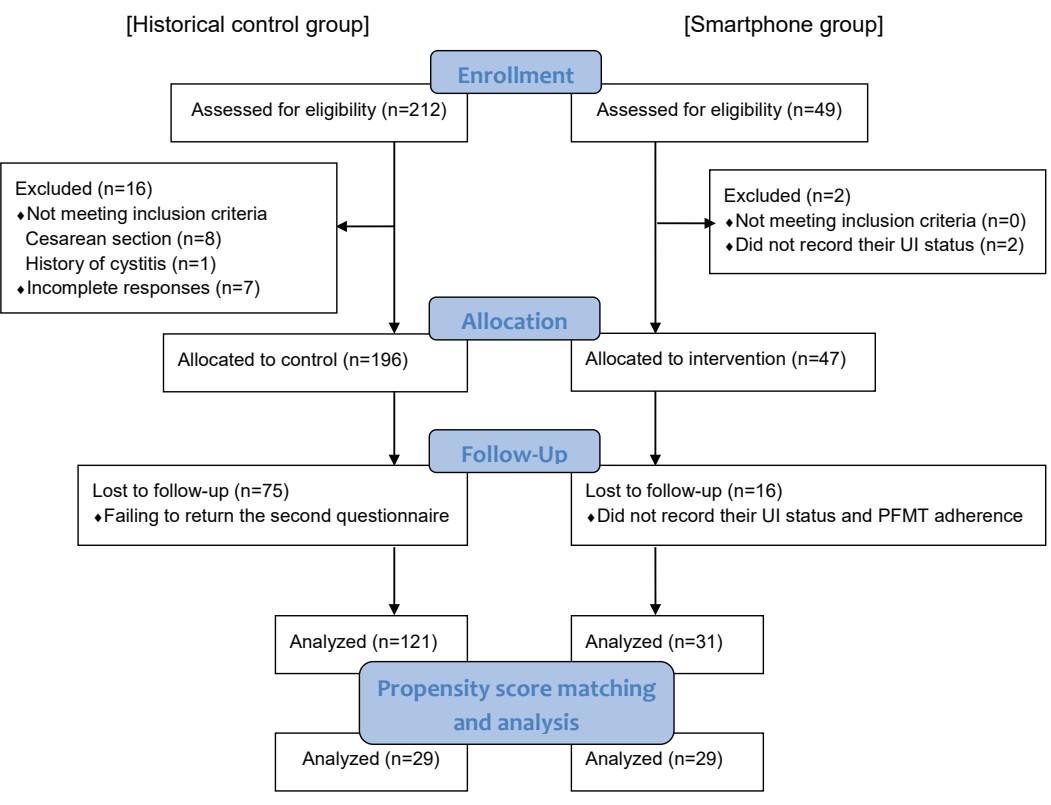

**Figure 2** Flow diagram of postpartum women participating in the study.

control group were included in the analysis. Propensity score matching (PSM) of the two groups resulted in 29 matched pairs. The groups' background characteristics before and after PSM are shown in Table 1. No significant differences in any background characteristics either before or after PSM were observed.

Table 2 shows the PFMT implementation rate for the eight-week intervention period, the PFMC reps/day and PFMT days/week of women who self-reported PFMT performance, and UI prevalence at baseline and at eight weeks. PFMT implementation rate was significantly higher in the smartphone group (69 vs. 31%, $p = 0.008$). Median values for PFMC reps/day and PFMT days/week were both significantly higher in the smartphone group (15 vs. 1 PFMC/day, $Z = -3.002$, $p = 0.006$) (7 vs. 3 days/week, $Z = -4.144$, $p < 0.001$). Among those who reported UI at baseline, three women in the smartphone group and three out of eight women in the control group experienced remission after eight weeks. In addition, two women had a newly developed UI in the control group, while no new cases were observed in the smartphone group. No difference in UI prevalence at baseline (10 vs. 2eight%, $p = 0.056$) was noted, but UI prevalence was significantly lower in the smartphone group at eight weeks (0 vs. 24%, $p = 0.004$). Compared with the control group, the smartphone group exhibited better PFMT adherence, in terms of PFMT implementation rate and median training intensity and training frequency. UI prevalence was not different between the groups at baseline, but was significantly reduced in the smartphone group at eight weeks.

**Table 1  Characteristics of smartphone and control groups before and after propensity score matching.**

| | Before PSM | | | After PSM | | |
|---|---|---|---|---|---|---|
| | Smartphone $n = 31$ | Control $n = 121$ | $p$ | Smartphone $n = 29$ | Control $n = 29$ | $p$ |
| Age (years) | 34 (31–36) | 33 (30–36) | 0.476[a] | 34 (31–36) | 34 (32–37) | 0.760[a] |
| Primipara | 7 (22.6) | 42 (34.7) | 0.281[b] | 5 (17.2) | 5 (17.2) | 1.000[b] |
| BMI before pregnancy (kg/m$^2$) | 19.7 (18.5–20.8) | 19.8 (18.6–21.0) | 0.671[a] | 19.7 (16.8–20.8) | 19.8 (18.5–21.2) | 0.732[a] |
| Weight gain during pregnancy (kg) | 10.0 (9.0–12.0) | 9.0 (7.4–11.0) | 0.067[b] | 10.0 (8.5–11.8) | 9.5 (8.0–12.0) | 0.691[a] |
| Birth weight (g) | 3,060 (2,749–3,456) | 3,150 (2,941–3,361) | 0.203[a] | 3,016 (2,736–3,349) | 3,214 (3,030–3,368) | 0.106[a] |

Notes.
Data are shown as median (interquartile range) or number (%).
PSM, propensity score matching; BMI, body mass index.
[a] Mann–Whitney U test.
[b] Fisher's exact test.

**Table 2  Comparison of urinary incontinence and pelvic floor muscle training adherence measures between smartphone and control groups after propensity score matching.**

| | Smartphone $n = 29$ | Control $n = 29$ | $p$ |
|---|---|---|---|
| PFMT adherence | | | |
|     Implementation rate | 20 (69.0) | 9 (31.0) | 0.008[a][*] |
|     Training intensity (PFMC reps/day) | 15 (3–18) | 1 (1–4) | 0.006[a][*] |
|     Training frequency (PFMT days/week) | 7 (6–7) | 3 (1–4) | <0.001[a][*] |
| UI prevalence | | | |
|     Baseline | 3 (10.3) | 8 (27.6) | 0.056[b] |
|     After 8 weeks | 0 (0.0) | 7 (24.1) | 0.004[b][*] |

Notes.
Data are shown as median (interquartile range) or number (%).
PFMT, pelvic floor muscle training; PFMC, pelvic floor muscle contraction; UI, urinary incontinence.
[a] Bonferroni correction followed by Mann–Whitney U test.
[b] Binomial test.
[*] Significant difference.

## DISCUSSION

In this study, postpartum women who received PFMT support via a smartphone-based reminder system had a higher PFMT implementation rate during the intervention period, higher PFMC reps/day, higher PFMT days/week, and improved UI prevalence after the eight-week intervention period compared with postpartum women who only received a verbal instruction for PFMT.

We adopted propensity score matching to compare observational data with historical controls. This technique allows us to analyze the data as a quasi-randomized controlled trial (*Qin et al., 2010*; *Haukoos & Lewis, 2015*). Our participants were not randomly selected. The intervention group consisted of women who consented to use the smartphone-based reminder system; thus, the risk of selection bias was unavoidable. In addition, the intervention group had few subjects. However, with PSM, we were able to minimize

problems related to cost and time and avoid the need for blinding, which is a difficult condition for PFMT interventions (*Qin et al., 2010*; *Haukoos & Lewis, 2015*). PSM allowed us to evaluate the PFMT intervention's effects on UI management using groups with uniform background characteristics, while adjusting for the confounding risk factors of age, parity, and BMI.

Any smartphone-based interventions have not been previously developed for UI. A previous study where women were coached by a physical therapist throughout four sessions of PFMT found that significantly more women in the intervention group performed PFMT in the previous month at 12 months after delivery (89 vs. 65% in control) (*Wilson & Herbison, 1998*). Another study performed an intervention where postpartum women received three PFMT instructional sessions at home administered by a midwife, health visitor, or continence adviser. The authors found that 12 months after delivery, women in the intervention group had significantly higher rates of performing PFMT in the previous month (79 vs. 48% in control), with significantly more PFMCs per day (20 vs. 5 in control) (*Glazener et al., 2001*). Our study demonstrated that PFMT adherence could be enhanced by simply adding a smartphone-based reminder system to a midwife-led PFMT instruction program, which has a similar efficacy to that of intensive PFMT protocols where women have to visit a medical facility for follow-up or are visited by a professional at home to receive coaching. Accordingly, the smartphone-based reminder system excels as a simple yet free way to assist postpartum women in performing PFMT in their free time at home. However, the withdrawal rate in the smartphone group was 36.7% (18/49). *Boyle et al. (2012)* reported that the withdrawal rate of intensive PFMT in antenatal and postnatal women was 4∼38%. We could not follow-up to clarify the reasons for withdrawal as there was no response, even when sending a message to record the UI status and PFMT adherence. Few studies have described the withdrawal rate of PFMT. In this way, our results are valuable as they provide important data on this point, but we must consider the possibility of detection bias.

*Morkved & Bo (1997)* also ran an eight-week intervention study on PFMT beginning after childbirth. Women in their intervention group received individual coaching from a physical therapist to ensure correct PFMC and received advice on how to continue the training at home. UI prevalence was not different between the intervention and control groups at baseline; after eight weeks, however, a significant between-group difference was observed (intervention, 14.4% [14/99]; control, 28.3% [22/99]). Apparently, for PFMT, both conventional intensive programs and our smartphone-based protocol could enhance adherence and lower UI occurrence. Moreover, our research findings suggested that the smartphone-based reminder system allows patients to successfully manage their UI, thereby helping them eliminate the symptoms and prevent its onset. By contrast, other studies reported that intensive PFMT interventions are ineffective at reducing UI incidence (*Sleep & Grant, 1987*; *Ewings et al., 2005*). A recent study investigated PFMT support provided by a physical therapist over the internet for three months. The study reported that while the intervention group perceived significant improvement in its condition and reported significantly reduced usage of incontinence aids compared with that at baseline, no significant differences were reported in the control group regarding the use of the postal

support alone (*Sjostrom et al., 2013*). One possible reason for such discrepancies could be the variable training contents and evaluation periods from study to study. Hence, more studies on PFMT with a careful selection of training contents and techniques for managing UI are necessary.

This study has two limitations. First, other factors that influence UI, PFMT implementation rate, and PFMT adherence may remain undiscovered. Thus, failure to adjust for unobserved confounders in the PSM model was possible. Hence, further clarification is necessary. Second, our research was a single-institution, non-randomized observational study. Our intervention's effectiveness needs to be further evaluated using a multi-center, large-scale research design.

The Japanese health care system does not offer rehabilitation aimed at recovering pelvic floor muscle performance in postpartum women. Postpartum women only receive verbal instruction regarding PFMT, which is the conservative treatment for urinary incontinence. Postpartum women develop UI from diminished pelvic floor muscle function following pregnancy and childbirth. Despite the fact that some women experience decreased HRQOL as a result (*Takaoka et al., 2017*), the country has not actively introduced health care reforms or offered related rehabilitation treatments, which could be attributed to costs and issues with ensuring adequate staffing. Our smartphone-based reminder system could eliminate these challenges and, if widely adopted, may improve the HRQOL of postpartum women with UI by enhancing PFMT adherence and improving their ability to self-manage the condition.

## CONCLUSIONS

Our smartphone-based reminder system appears promising in enhancing PFMT adherence and managing postpartum UI in postpartum women. By enhancing PFMT adherence and improving women's ability to manage the condition, the reminder system could improve the health-related quality of life of postpartum women with UI.

## ACKNOWLEDGEMENTS

We are grateful to all the women who participated in the study. We also thank Dr. Hiroaki Jikihara, Director of Jikihara Women's Clinic; Chief Nurse Kayoko Furusawa; and the clinic's midwives.

### Funding

This project was supported by a Grant-in-Aid for Young Scientists (B), grant No. A227922240, from the Ministry of Education, Culture, Sports, Science and Technology. There was no additional external funding received for this study. The funders had no role in study design, data collection and analysis, decision to publish, or preparation of the manuscript.

## Grant Disclosures

The following grant information was disclosed by the authors:
The Ministry of Education, Culture, Sports, Science and Technology: A227922240.

## Competing Interests

The authors declare there are no competing interests.

## Author Contributions

- Kaori Kinouchi conceived and designed the experiments, performed the experiments, analyzed the data, contributed reagents/materials/analysis tools, wrote the paper, prepared figures and/or tables, reviewed drafts of the paper.
- Kazutomo Ohashi conceived and designed the experiments, analyzed the data, reviewed drafts of the paper.

## Human Ethics

The following information was supplied relating to ethical approvals (i.e., approving body and any reference numbers):

This research was approved by the Osaka University Medical Science Department of Health Ethics Committee.

## Data Availability

The raw data has been provided as a Supplemental File.

## Supplemental Information

Supplemental information for this article can be found online at http://dx.doi.org/10.7717/peerj.4372#supplemental-information.

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
