# Peer review of "Smartphone-based reminder system to promote pelvic floor muscle training for the management of postnatal urinary incontinence: historical control study with propensity score-matched analysis"

_PeerJ, doi:10.7717/peerj.4372_

## Round 0.1 · original submission · Minor Revisions

Please address all the reviewer comments in a rebuttal letter indicating each comment and how you addressed it.Thank you.

·

Basic reporting

a. Line 16 the reference is listed as 1, and the rest of the paper is using author last names.
b. Clear/English
i. Line 43, I think you're trying to compare postpartum women to postmenopausal women? It may be more clear to state the group you are comparing them with. After all, postpartum women are non-pregnant women.
ii. Line 86, I suggest "All data" instead of "All the data"
iii. Line 113, Should it say "this data is not published"?
iv. Line 118/119, I think you could say "The participant who agree to participate answered a similar initial questionnaire to the experimental group for background characteristics and current UI status."
v. Line 185, the "two and 16" is confusing. Are these the participant numbers? If so, it isn't necessary and you could just state that two participants were excluded and the reason.
c. References/sufficient background and context-Yes
d. Raw data shared-Yes, I am able to open and view this file.

Experimental design

a. Original-Yes
b. Aime and scope-The focused aim for this study is a strength.
c. Question well defined and meaningful-Yes, I believe it is interesting and meaningful research.
d. Identified knowledge gap-There is a clear gap between the knowledge and the action of the women with UI. This study looks at this gap.
e. Method described in detail-Yes, I believe this study could be replicated based on the description in the article.

Validity of the findings

Impact of this study was assessed. I believe this study could be replicated based on the description in the article.

Conclusion is well stated and linked to the original research question.

Additional comments

a. In line 114, I think it would be nice to either have the same exclusion criteria or list the exclusion criteria for the control group so the reader knows there wasn't bias toward one group over the other.

Reviewer 2 ·

Basic reporting

Thank you for the opportunity to review this article. The article provides sound findings regarding the effectiveness of the smartphone-based reminder system to enhance PFMT adherence and to manage UI in postpartum women. However, there are some issues that need improvement. These issues are outlined below:

Major Revision

I wonder about the withdrawal. You describe that the smartphone-based reminder system excel as a simple free way to assist postpartum women in performing PFMT in their free time. But sixteen women in the smartphone group did not record their current UI status after 8 weeks. It is 32.7% of enrollments. Why did not you discuss this point.

Minor Revisions

1. Line 16
loss of urine.”1) In a population   
Please delete 1).

2. Line 56-58
I could not understand this sentence. Why do you put a parentheses.

3. Line 79
Is ICI a mistake of ICS?

4. Line 101
Who is a midwife? Does the midwife belong to the clinic?

5. Figure 2
Historical control group and Smartphone group are assigned from a beginning in this research. I think that you should show a group name the upper part of the figure.

6. Figure 2, Follow-Up section
In the Historical control group, A parenthesis is not written.

Experimental design

Please provide more information about the exclusion criteria in the control group in the Materials & Methods section. You describe that it is similar to those in the smartphone group. It isn’t same. Please explain what is different.

Validity of the findings

no comment

Additional comments

no comment

---

## Round 0.2 · accepted · Accept

Thank you very much for addressing the reviewers' comments. I am pleased to tell you your manuscript has been accepted for publication with PeerJ.

·

Basic reporting

Line 91, I think the parenthesis within the parenthesis should be brackets.

Experimental design

No comment.

Validity of the findings

No comment.

Additional comments

This version of your manuscript is more clear and concise. Thank you for the opportunity to review your work.